# Recombinant Antigen of Type 2 Porcine Reproductive and Respiratory Syndrome Virus (PRRSV-2) Promotes M1 Repolarization of Porcine Alveolar Macrophages and Th1 Type Response

**DOI:** 10.3390/vaccines9091009

**Published:** 2021-09-10

**Authors:** Rika Wahyuningtyas, Yin-Siew Lai, Mei-Li Wu, Hsin-Wei Chen, Wen-Bin Chung, Hso-Chi Chaung, Ko-Tung Chang

**Affiliations:** 1Research Centre for Animal Biologics, National Pingtung University of Science and Technology, Neipu, Pingtung 912, Taiwan; rikawahyurizky@gmail.com (R.W.); rindy.0802@gmail.com (Y.-S.L.); mlwu@mail.npust.edu.tw (M.-L.W.); 2Department of Veterinary Medicine, National Pingtung University of Science and Technology, Neipu, Pingtung 912, Taiwan; wbchung@mail.npust.edu.tw; 3Department of Food Science, National Pingtung University of Science and Technology, Neipu, Pingtung 912, Taiwan; 4National Institute of Infectious Diseases and Vaccinology, National Health Research Institutes, Miaoli 350, Taiwan; chenhw@nhri.org.tw; 5Graduate Institute of Biomedical Sciences, China Medical University, Taichung 400, Taiwan; 6Graduate Institute of Medicine, College of Medicine, Kaohsiung Medical University, Kaohsiung 800, Taiwan; 7Flow Cytometry Center, Precision Instruments Center, National Pingtung University of Science and Technology, Neipu, Pingtung 912, Taiwan; 8Department of Biological Science and Technology, National Pingtung University of Science and Technology, Neipu, Pingtung 912, Taiwan

**Keywords:** PAMs, CD163, PRRSV, M1, M2, Th1

## Abstract

The polarization status of porcine alveolar macrophages (PAMs) determines the infectivity of porcine reproductive and respiratory syndrome virus (PRRSV). PRRSV infection skews macrophage polarization toward an M2 phenotype, followed by T-cells inactivation. CD163, one of the scavenger receptors of M2 macrophages, has been described as a putative receptor for PRRSV. In this study, we examined two types of PRRSV-2-derived recombinant antigens, A1 (g6Ld10T) and A2 (lipo-M5Nt), for their ability to mediate PAM polarization and T helper (Th1) response. A1 and A2 were composed of different combination of *ORF5*, *ORF6*, and *ORF7* in full or partial length. To enhance the adaptive immunity, they were conjugated with T cells epitopes or lipidated elements, respectively. Our results showed that CD163^+^ expression on PAMs significantly decreased after being challenged with A1 but not A2, followed by a significant increase in pro-inflammatory genes (*TNF-α*, *IL-6*, and *IL-12*). In addition, next generation sequencing (NGS) data show an increase in T-cell receptor signaling in PAMs challenged with A1. Using a co-culture system, PAMs challenged with A1 can induce Th1 activation by boosting IFN-γ and *IL-12* secretion and *TNF-α* expression. In terms of innate and T-cell-mediated immunity, we conclude that A1 is regarded as a potential vaccine for immunization against PRRSV infection due to its ability to reverse the polarization status of PAMs toward pro-inflammatory phenotypes, which in turn reduces CD163 expression for viral entry and increases immunomodulation for Th1-type response.

## 1. Introduction

Porcine reproductive and respiratory syndrome virus (PRRSV) causes severe respiratory and reproductive disease in swine worldwide. According to the current genetic analysis by the International Committee on Taxonomy of Viruses (ICTV) PRRSV has been reclassified to the Betaarterivirus genus, belonging to the order Nidovirales and the Arteriviridae family. Type 1 (European) and type 2 (American) are the two main genotypes of PRRSV, which have been classified into two distinct virus species (namely Betaarterivirus suid 1 and Betaarterivirus suid 2) [1]. European PRRSV has further divided into three subtypes, pan-European subtype 1 and East European subtypes 2 and 3, and at least nine different genetic lineages of North American PRRSV have been classified [2]. It consists of an enveloped virus, positive single-stranded RNA and is encapsulated in nucleocapsid proteins containing 11 open reading frames (*ORFs*), identified as *ORF1a*, *ORF1b*, *ORF2a*, *ORF2b* and *ORF3-7* [3,4,5,6]. *ORF5*, *ORF6* and *ORF7* encode a glycosylated membrane protein GP5, a non-glycosylated membrane protein M, and the nucleocapsid protein N [7,8].

Receptor-mediated endocytosis and replication are processes of PRRSV entry into host cells [9]. Here, there are six potential host cell receptors for PRRSV attachment, namely heparin sulphate, vimentin, CD151, CD163 (scavenger receptor for the haemoglobin-haptoglobin complex), sialoadhesin (CD169), and DC-SIGN (dendritic cell-specific intercellular adhesion molecule-3-grabbing non-integrin, known as CD209) [10]. GP5 is the dominant glycoprotein on the viral surface and is considered the major envelope glycoprotein. In contrast, the glycoproteins GP2, GP3, and GP4 were recognized as the minor envelope glycoproteins due to their lower presence on the viral surface [8]. Nowadays, *ORF5* is being one of the most promising targets for the development of the new generation of vaccines against PRRSV. However, some of the experimentally developed PRRSV-expressing GP5 vaccines showed that the neutralizing antibodies are weak [11]. In Wang’s (2009) study [12], they immunized mice and pigs with recombinant adenovirus expressing porcine GM-CSF-GP3-GP5 (rAd-GF35) and determined the responses after immunological challenges. The results described that the antibodies against PRRSV GP3/GP5 developed in animals after inoculation with rAd-GF35. Nonetheless, they did not find the secretion of IFN-γ after challenge and they did not measure the correlation between the level of cell-mediated immunity (CMI) and viremia. IFN-γ and *TNF-α* are important cytokines that induce the response of CD4^+^ T-cells upon vaccination [13]. However, the underlying mechanism of T-cells-mediated immunity to PRRSV is not fully understood and further studies are needed.

Currently, the most efficient and functional method to prevent the PRRSV disease is vaccination. The commercially attenuated live PRRSV vaccines are a tool for valuable disease control, but have limited efficacy in protecting against infections with the genetically diverse field strains of PRRSV and carry the potential risk of becoming virulent again. Lower efficacy in preventing infectious disease has also been observed with killed PRRSV vaccines [14,15,16,17]. To date, the success of the majority of vaccines relies on their capability to target specific pathogen antigens which could induce humoral immune response in the host. On the other hand, cell-mediated immunity is less achievable with current vaccination strategies, and this leads to most chronic infections. Some developed vaccines carry several potential risks, such as the risk of co-purification of undesirable contaminants or reversal of toxoids to their toxigenic form when considering diphtheria or tetanus toxoid vaccines. Recombinant protein vaccines have the potential to avoid these potential risks [18]. B- and T-cells have become the focus of vaccine development research in recent years. However, innate immune cells have a pivotal role in vaccination to promote the expression of long-lasting adaptive immune responses. In order to get an effective vaccination, we need an immune response to immunization, and this can be provided by both the innate and adaptive immune systems. Furthermore, there are two critical points to be considered in order to produce an effective immunization, which are the induction of long-term stimulation of the humoral and cell-mediated arms of the adaptive system that can be produced by the production of effector cells and memory cells [19]. Salerno et al. (2012) [20] reported that the induction of strong and persistent memory T-cells and cell-mediated immunity (CMI) responses supported vaccination success. CMI responses were dependent on two cell types, namely CD4^+^ and CD8^+^ T-cells. CD8^+^ T-cells protecting the host cells from pathogen by destroying infected or tumor cells and secreting interferon-γ (IFN-γ) and other cytokines. CD4^+^ T-cells play the important role in host defense supported by the two main subpopulations: Th1, which in generally trigger inflammatory responses and the differentiation of CD8^+^ cells, and Th2, which support the production of specific antibodies.

Primary porcine alveolar macrophages (PAMs) are the major target host cells of PRRSV infection and are known to express CD14, SLA II, CD163, CD169, CD203, SWC3 (CD172a), and CD16 receptors [21,22]. A previous study demonstrated that by introducing the *CD163* gene alone into immortalized PAMs (iPAMs) was sufficient to restore PRRSV susceptibility, suggesting that CD163 was the sole determining factor in PAMs for PRRSV entry [23]. CD163 is a scavenger receptor expressed on monocytes/macrophages and was determined to be a well-characterized M2 marker since its expression was upregulated during macrophage differentiation of human blood monocytes stimulated by macrophage colony-stimulating factor (M-CSF) [24,25]. Moreover, CD163 reduced the expression of *TNF-α* and other pro-inflammatory factors [26]. High expression of CD163 in macrophages indicates potent anti-inflammatory potential and phagocytic ability of macrophages to clear debris and apoptotic cells [25]. Wang (2017) [27] demonstrated that the M1 type strongly inhibits replication of highly pathogenic PRRSV (HP-PRRSV), but not in M2-type PAMs. Moreover, their results showed that HP-PRRSV infection promotes the repolarization of M2-type PAMs. These findings strongly suggested that HP-PRRSV infection can modulate macrophages polarization. The presence of CD163^+^ in M2-type macrophages suppresses T-cell proliferation and pro-inflammatory cytokine secretion. On the contrary, T-cells proliferation and pro-inflammatory cytokine production was significantly increased in CD163-deficient macrophages in osteosarcoma [28]. M2 macrophage polarization was mediated by IL-4 alters secretion of IL-13 during Th2-type response. In contrast, the first line of defense against intracellular pathogens was initiated by M1 macrophages through the secretion of *IL-12*, which promoted or enhanced the Th1 type of CD4^+^ lymphocytes [29]. Th1 cells also provoke the production of immunoglobulin G2a (IgG2a) antibodies in B-cells to optimize the ability of viral clearance and to clear extracellular bacteria [30]. In our previous study, the production of neutralizing antibodies in vaccinated pigs was generally poor in parallel with showing an immunosuppressive serum by increasing IL-10 and Treg cells. Therefore, recombinant antigen derived from PRRSV was thought to induce immunosuppression, just as a true virus, through the induction of M2 polarization of PAMs in pigs. The aim of this study was to evaluate the effects of the two recombinant antigens, A1 (g6Ld10T) and A2 (lipo-M5Nt), with respect to their ability to mediate innate and T-cell-mediated immunity prior to humoral immunity after vaccination, as a novel vaccine candidate.

## 2. Materials and Methods

### 2.1. Ethics Statement

The lung collection and pig euthanization were approved by the Institutional Animal Care and Use Committee (IACUC) of Veterinary Medicine at National Pingtung University of Science and Technology (NPUST), Taiwan.

### 2.2. Pigs and Inoculations

This study was performed on specific pathogen free (SPF) piglets, approximately eight to eleven weeks of age and nine to twelve kg in weight. They were raised in a room with positive pressure in NPUST Animal Diagnostic Center. 

### 2.3. Constructed the Recombinant Protein Antigen

Antigen 1 (A1) was constructed from the complete sequence of *ORF5* combined with the partial sequence of *ORF6* and the copies of T-cell epitopes (Appendix A). The recombinant antigen was expressed using the baculovirus expression system. Lipidated recombinant protein antigen 2 (A2) was constructed from the combination of the *ORF5*, *ORF6* and *ORF7* sequences and expressed using the *Escherichia coli* (*E. coli*) expression system (Appendix A).

### 2.4. Collecting Porcine Alveolar Macrophages (PAMs)

Pigs were euthanized by exsanguination. The trachea was ligated to prevent total pulmonary collapse, followed by the removal of heart and lungs from the thorax. Alveolar macrophages were harvested aseptically from fresh lungs. The lungs were washed intratracheally 2–4 times with phosphate buffered saline (PBS), and the wash fluids containing PAMs were centrifuged for 10 min at 800× *g*. The collected PAMs were seeded in 12-well plates and maintaining with complete RPMI-1640 medium (Corning, Manassas, VA, USA) containing 10% fetal bovine serum (FBS) (Hyclone, Logan, UT, USA) at 37 °C in a humidified 5% CO_2_ atmosphere. 

### 2.5. Cytokine Stimulation

PAMs were seeded in 12-well plates and divided into five groups; control (untreated); 1 µg/mL LPS (Sigma-Aldrich, Steinheim, Germany) added; 20 ng/mL IL-4 (BIOTECH, INC, Alpharetta, GA, USA) added; 5 μg/mL of A1 added; and 5 μg/mL of A2 added. After 24 h of treatment, cells were collected to extract RNA and perform qPCR analyses.

### 2.6. RNA Extraction and Quantitative Real-Time Polymerase Chain Reaction

Total RNA was prepared using Trizol reagent (Invitrogen, Waltham, MA, USA) according to the manufacture’s protocol. Total RNA (1 μg) was used in the reverse transcription (RT) reaction by iScript cDNA Synthesis Kit (Bio-Rad, Hercules, CA, USA). The quantitative real time PCR was performed using KAPA SYBR^®^ FAST qPCR Master Mix (2X) Kit (KAPA Biosystem, Wilmington, DE, USA) according to the manufacturer’s protocol. Quantitative PCR reactions were performed using a QIAGEN Rotor Gene Q Real-Time PCR. *β-actin* was used as the endogenous reference gene since it has the highest stability across pig tissues compared to the other reference genes [31]. The primer sequences (5′–3′; forward, reverse) were showed in Table 1. The amplification steps were set for 3 min at 95 °C, followed by 40 cycles of denaturation at 95 °C for 3 s, and annealing at 60 °C for 20 s. The data were calculated using the standardized mRNA level comparative methods 2^−ΔΔCt^. The 2^−ΔΔCt^ method is a convenient way to analyze the relative changes in gene expression with a high efficiency qPCR assay [32].

### 2.7. Next Generation Sequencing (NGS) Analysis

The cDNA libraries were collected from pooling of six independent samples in each group. Then, cDNA libraries were assessed on the Agilent Bioanalyzer 2100 system and a Real-Time PCR system. Housekeeping gene (*β-actin*) was served as an internal control to verify the cDNA quality and quantity in PCR. NGS was performed externally at the Genomics on an Illumina Novaseq 6000 with 150 bp paired-end reads (Genomics, BioSci & Tech Company, New Taipei City, Taiwan). Raw-sequencing reads were filtered using the program Trimmomatic (version 0.36) [33]. Read alignments were assembled using Bowtie2 (version 2.3.5) [34]. The raw gene counts were extracted with RSEM (version 1.3.3) [35]. The R package EdgeR v3.16.5 tool was used for differential gene expression analysis between two sample groups. In addition, the log_2_ Fold Change (log_2_ FC) was calculated using the log_2_ (sample count A/sample count B). Significant differences between two sets of samples were identified by t-test with *p* < 0.05. A gene ontology (GO) enrichment analysis was conducted on the differential genes obtained through screening, and when *p* < 0.05, the GO terminology is regarded as significantly enriched [36]. The Kyoto Encyclopedia of Genes and Genomes (KEGG, https://www.genome.jp/kegg/kegg2.html (accessed on 26 January 2021). was used for gene enrichment of differentially expressed genes [37]. The datasets presented in this study can be found in online repositories (NCBI Bioproject PRJNA665327:https://dataview.ncbi.nlm.nih.gov/object/PRJNA726625) (accessed on 1 May 2021).

### 2.8. Integration of the Protein-Protein Interaction (PPI) Network

The potential differential expression gene (DEG) interactions at the protein level were explored by the Search Tool for the Retrieval of Interacting Genes (STRING; string-db.org). The PPI networks of DEGs by STRING were derived from validated experiments. *p* < 0.05 was considered to indicate a statistically significant difference.

### 2.9. Isolation of Porcine Peripheral Blood Mononuclear Cells (PBMC) 

Peripheral blood mononuclear cells (PBMCs) were isolated by using Ficoll-Paque (GE Healthcare BioScience, Uppsala, Sweden) density-gradient centrifugation at 400× *g* for 30 min according to the manufacturer’s instructions. PBMCs were washed three times in RPMI 1640 (Corning, Manassas, VA, USA), and resuspended in advanced RPMI 1640 medium containing 10% fetal bovine serum (FBS) (Hyclone, Lo-gan, UT, USA) for further experiments. 

### 2.10. Fluorescence Activated Cell Sorting (FACS) for T-Cell Subsets 

For T-cells subset, PBMCs were sorted by FACS into subpopulations for co-culture experiments. Briefly, 1 × 10^6^ cells/mL were stained with 0.05 µg anti-pig FITC-CD4^+^ antibody (Ab24989, Abcam, Cambridge, UK) and 10 µg anti-pig CD25^+^ primary antibody (MCA1736GA, Bio-Rad) on ice in dark for 30 min, followed by staining with 5 µg anti-mouse R-Phycoerythrin-conjugated IgG1 secondary antibody (STAR132PE, Bio-Rad) on ice in dark for 30 min. Hereafter, the stained cells were washed twice with cold PBS before acquisition on BD FACS Aria II cells sorter (BD Biosciences, San Jose, CA, USA). Two subpopulations of T cells were sorted based on the expression of CD4^+^CD25^−^ (Th1) and CD4^+^CD25^+^ (Treg). 

### 2.11. Flow Cytometry Analysis

PAMs (1 × 10^6^ cells/mL) were collected and washed once with cold PBS containing 0.5% BSA (Sigma-Aldrich, Steinheim, Germany). Cells were incubated with fluorescence-conjugated antibodies in 1 µg FITC-SLA II^+^ (Bio-Rad), 5 µg FITC-CD14^+^ (Invitrogen), 0.1 µg APC-CD80^+^ (Invitrogen), and 1:10 dilution of PE-CD163^+^ (Invitrogen), on ice in dark for 30 min. The cells were then washed by centrifugation at 300× *g* for 5 min and resuspended in 500 μL to 1 mL of cold PBS. The expression of surface protein on cells were measured by using the flow cytometry (BD Biosciences, San Jose, CA, USA), and the analysis was performed using BD FACSDiva Software (BD Biosciences) and FlowJo Software (Tree Star, Inc., Ashland, OR, USA).

### 2.12. Th1 Cytokines Assay

In an indirect transwell co-culture system, 2 × 10^6^ PAMs were seeded at the bottom of the transwell system overnight. Then after, the cells were stimulated by antigens A1 or A2 and 2 × 10^5^ T-cells subset (CD4^+^CD25^−^ or CD4^+^CD25^+^) were seeded into the upper chamber of the transwell insert, simultaneously. The co-culture dish was followed by incubation in standard conditions (5% CO_2_; 37 °C) for 48 h. Then, conditioned medium was collected for Th1 cytokines assay. Briefly, cytokine concentrations were determined from a standard curve created by a reference preparation of IL-10, IFN-γ (Thermo Fisher Scientific, Vienna, Austria) and *IL-12* (R&D System, Abingdon, United Kingdom) from commercial ELISA kit according to the protocols provided by the manufacturers. The optical density A450 nm of each well was measured by EZ Read 400 Microplate Reader (Biochrom, Cambridge, UK).

### 2.13. Statistical Analysis

Data were presented as mean ± SEM when indicated. Statistical analysis was conducted by *t*-test with a 95% confidence limit, and one-way ANOVA followed by Tukey’s test for multi comparisons based on a Shapiro–Wilk test for normality. The data analysis was performed using Prism 8.0 (GraphPad Software Inc., San Diego, CA, USA). Differences were considered significant at *p* < 0.05. 

## 3. Results

### 3.1. A1 Directs Macrophages Polarization toward M1 Macrophages and Downregulation of CD163 Expression

A homogeneous and stable subset of cells compatible with PAMs due to their size and granularity were identified in all control and PRRSV-infected pigs. However, this subset decreased proportionally in infected groups compared to the control. We presented the gating strategy to obtain the living cells in terms of cell size and granularity based on forward light scatter/side light scatter (FSC-A/SSC-A) and excluded the aggregated cells by FSC-W/FSC-H as well as SSC-W/SSC-H (Figure 1a). Since PAMs represent the PRRSV target cells in pigs, the first objective of our study was to determine the specific markers associated with these cells. Among them, CD14 was first investigated as a marker of the myelomonocytic phenotype. A representative histogram is shown in Figure 1b–d. Furthermore, PAMs are known to express both SLA II and CD163 receptors [38]. In agreement with the previous literature, we confirmed that 91.3%, 97.5%, and 95.2% PAMs isolated from healthy pigs expressing CD14, SLA II, CD163 positive, respectively. On the contrary, only a small percentage of cells (23.8% and 23.1%) expressing CD14^+^ and CD163^+^ were found in PAMs isolated from PRRSV-infected pigs. This result complements an article which demonstrated that the proportion of CD163^+^ cells was decreased after eight days post-infection of PRRSV [39]. Meanwhile, our result also showed a significant decrease in the percentage of cells expressing CD14^+^, SLA II^+^, and CD163^+^ between PRRSV-infected and healthy pigs (Figure 1e). Nevertheless, the mean fluorescence intensity (MFI) of CD14^+^ and CD163^+^ was significantly higher in PAMs isolated from PRRSV-infected pigs. The increase in CD14^+^ indicates a strong infiltration of monocytes in the lungs, which is in agreement with the statement of Van., et al. (2004) [40], who showed that CD14 was significant increase in the lungs of pigs after PRRSV infection. A significant increase in MFI of CD163^+^ during late gestation and PRRSV type 2 infection in pregnant gilts was also demonstrated in the previous study by Novakovic, et al. (2016) [41]. Moreover, the significant increase in MFI of CD163^+^ in PRRSV-infected pigs probably was due to the polarization of the M2 phenotype after infection, and indicates that CD163 plays an important role in receptor for viral infection (Figure 1f).

To further validate the effect of antigens on macrophages polarization, we challenged PAMs with recombinant PRRSV-2 antigens (A1 and A2). The results showed that PAMs challenged with A1 enhanced the upregulation of pro-inflammatory genes (*TNF-α*, *IL-6* and *IL-12*), suggesting that A1 promotes M1 macrophages polarization, but not in PAMs challenged with A2. In contrast, A2-challenged PAMs up-regulate the expression of *Arg-1*, one of the anti-inflammatory gene representatives for M2 macrophages (Figure 2a). These results suggest that A1 may be a candidate for inhibiting PRRSV infection by regulating macrophages polarization. There were no significant differences on SLA II^+^ and CD80^+^ surface protein marker between A1 and A2 induction. However, the number of cells-expressing CD163^+^ were significantly decreased after they were treated with A1 (Figure 2b).

### 3.2. A1 Stimulate Endogenous Pro-Inflammatory Gene for T-Cell Receptor (TCR) Signaling Pathway

To verify additional pro-inflammatory genes involved in the regulation of macrophages polarization by these recombinant antigens, we next performed transcriptome analysis of PAMs challenged with A1 versus A2. Our results showed that some of the pro-inflammatory genes, such as *Nf-kB*, *NNT*, *TNF-α*, and *JARID2*, were included in the top 50 upregulated genes in PAMs challenged by A1 (Figure 3a). In contrast, some of anti-inflammatory genes, such as *Arg-1*, *SLC7A6*, *MAP4*, and *GATAD2A*, were included in total top 50 downregulated genes (Figure 3b). Some of the differential expression genes also showed the significant changes in A1 compared to control (shown in the highlighted bars). We interestingly found that several special genes have different variants in the heatmap. These genes with the variants were represented by the code number on the Y-axis of the heatmap (Appendix A). Except for this, the majority of anti-inflammatory genes among the variants were downregulated in A1 compared to A2, such as *CD274* and *UBAP2L*. In contrast, the majority of *TLR8* variants, one of inducer of IFN-γ, were up-regulated in A1 compared to that in A2. In order to verify the quality and quantity of cDNA libraries in the PCR conditions, we used the *β-actin* as an internal control. Our results showed that the melting curve of *β-actin* reflects the specific amplicon in RT-PCR (Appendix A). In addition, the cycle threshold (Ct) values of *β-actin* in all samples among the groups were in consistency, by a range of 19–24, showing no significant differences statistically (Appendix A). Our *β-actin* gene expression level was also elucidated from the NGS data, which also manifested no significant differences among the groups by log_2_ FC value (Appendix A). According to these aforementioned results, the quality and quantity of cDNA libraries were identical among the groups. There were some overlapping genes among the A1 and A2 (Figure 3c). Furthermore, the data showed that there were 8 overlapping pathways enriched in up- and down-regulated genes. Nevertheless, T-cell receptors (TCRs) signaling pathway was enriched only in up-regulated genes, which were promoted by A1. This finding strongly suggests that endogenous genes expressed in A1-challenged PAMs promote the activation of TCR signaling pathway and regulate the immune system process (Figure 3d,e).

### 3.3. A1 Potentially Regulate Immune Response by Promoting Rap1 Signaling Pathway and Protein-Protein Interaction (PPI) Network

Furthermore, we investigated the correlating genes involved in the immune response process in their respective signaling pathways using the Kyoto Encyclopedia of Genes and Genomes (KEGG) enrichment analyses mapping tool. Surprisingly, we found that the Rap1 pathway, the key regulator of T-cells activation, was highly enriched in PAMs induced by A1 (Figure 4a). Consistent with our previous finding, the Rap1 pathway could be stimulated by TCR activation, leading to proliferation, survival, and gene activation (Figure 4b). Similarly, we found that the C-type lectin receptors (CLRs) signaling pathway were also enriched. CLRs promote various signaling pathways that lead to the expression of specific cytokines that determine T-cell differentiation status [42]. To investigate the potential protein-level interactions of differentially expressed genes (DEG) in regulating the immune response, we generated an analysis of predicted protein-protein interactions (PPI) of differentially expressed genes using Search Tool for Retrieval of Interacting Genes (STRING) analysis. The results showed that some of the predicted proteins secreted by PAMs challenged by A1 were correlated with inflammatory and immune responses, such as caspase1, IL-18, PIK3CB, and IKKB (Figure 4c).

On the other hand, downregulation of anti-inflammatory genes could act to suppress estrogen signaling, as in the KEGG mapping tool (Figure 5a,b). Additionally, these genes likely stimulate the downregulation of three negative regulator proteins of T-cell activation, namely ISG15, USP18, and DAPK1, as shown in Figure 5c.

### 3.4. PAMs Induced by A1 Activate Th1-Cells by Boosting Th1 Cytokine Secretion and TNF-α Expression 

It is generally believed that T helper (Th1) and regulatory T-cells (Treg) play an important role in dampening immune responses to antigens. To determine whether PAMs challenged by A1 are able to induce significant modulations in the mRNA level of transcription factors and cytokines related to the differential polarization of the T-cells immune response (Th1 and Treg), we observed the cytokine secretion profile of IL-10, *IL-12*, and IFN-γ using ELISA. The results showed that there was no significant difference in the secretion of IL-10 (Figure 6a). Remarkably, A1 induced the secretion of IFN-γ from PAMs co-cultured with Th1 cells (Figure 6b), representing a Th1 cells activation. Recently, Tregs have been shown to inhibit the proliferation of several immune cells, including B-cells, natural killer (NK) cells, natural killer T (NKT) cells, CD4^+^, and CD8^+^ T-cells, as well as monocytes and dendritic cells (DCs) [43]. Additionally, *IL-12* was released from both Th1/Treg cells triggered by A1 (Figure 6c). Related to these results, real-time quantitative RT-PCR analysis showed that A1-challenged PAMs cocultured with Th1 up-regulated the expression of *IL-6* (Figure 6d). Interestingly, cell-cell interaction of A1-induced PAMs with T cells increased *TNF-α* expression up to 80-fold (Figure 6e), compared with A1-induced PAMs alone in monoculture system, as shown in our previous result (Figure 2a). However, there were no significant differences in *Arg-1* expression (Figure 6f). We emphasize that the Th1-type immune response stimulated by A1 is purely derived from A1 protein itself, since the baculovirus expression system has advantages of free endotoxin in recombinant protein production compared to that from the *E. coli* expression system we used for A2 production. Cox (2009) [44] have shown that the baculovirus expression vector system produces large amounts of proteins that have similar biological activities to the original proteins than proteins expressed in bacterial systems. In addition, the use of insect cells can help to eliminate endotoxin contamination, leads to endotoxin-free recombinant antigen, and stimulates cytokine secretions. In contrast, the *E. coli* expression system has several disadvantages, including the LPS endotoxin [45]. LPS contains lipid A, a non-repeating “core” of oligosaccharide that has the endotoxic properties and is recognized by its Toll Like Receptor 4 (TLR4). It was known that TLR4 is expressed on the cell surface of macrophages and the activation of TLR4 mediates the inflammatory response in lung macrophages [46].

## 4. Discussion

PRRSV infection was indicated by pregnancy loss and poor reproductive performance in pregnant sows, while in young pigs and in piglets it showed in symptoms such as respiratory distress and high excess mortality [47]. In PRRSV, the major envelope glycoprotein (GP5) encoded by *ORF5* forms a heterodimer membrane protein with non-glycosylated M protein encoded by *ORF6*. *ORF7* encodes the nucleocapsid (N) protein and is known to be highly immunogenic in infected animals [48]. Here, we observed the two types of recombinant proteins mainly derived from antigenic epitopes of PRRSV-2 in stimulating innate and T-cell-mediated immunity as a novel vaccine candidate against PRRSV. A1 was designed from the complete sequence of *ORF5* in combination with a partial sequence of *ORF6* and provided with copies of T-cell epitopes so that it was delivered by the baculovirus expression system. Sewel et al., 2020, [49] showed that T-cells play an important role in protecting against many viral infections through processes known as cellular immunity. A2 was constructed as a lipidated recombinant protein containing *ORF5*, *ORF6* and *ORF7* generated using the *E. coli* expression system. 

PRRSV specifically infects certain subsets of differentiated macrophages in the lung and replicates in monocytic lineage cells, particularly porcine alveolar macrophages (PAMs) [27]. Macrophages play an important role in immune effectors and antigen presenting cells [50]. According to their functions and expression markers, macrophages were classified into classically activated macrophages (M1/kill macrophages) and alternatively activated macrophages (M2/repair macrophages) [51,52,53]. M1 macrophages were characterized by high expression levels of the major histocompatibility complex class II (MHC II), the CD68^+^ marker, and the co-stimulatory molecules CD80^+^ and CD86^+^. In addition, upregulation of the intracellular protein Suppressor of Cytokine Signaling 3 (SOCS3) and activation of inducible nitric oxide synthase (NOS2 or iNOS) have been demonstrated in M1 macrophages. In contrast, activation of M2 macrophages results in the secretion of high amounts of IL-10 and low amounts of *IL-12*. To promote their growth, they have also been shown to express high amounts of E- and C-type scavenger mannose and galactose receptors [54]. The expression of some surface markers such as mannitol receptor, CD206, CD163, CD209, FIZZ1, and Ym1/2 were also used to identify M2 macrophages [55].

In terms of mean fluorescence intensity (MFI), the expression of CD163 was significantly up-regulated on PAMs infected with PRRSV, but not with A1 (Figure 1f and Figure 2b). However, the total percentage of CD163^+^ cells was significantly decreased after A1 challenge (Figure 2b). A study proved that CD163 is critical for PRRSV replication and plays an important role in mediating viral internalization and degradation. Overexpression of CD163 contributes to the susceptibility of non-permissive cell lines to PRRSV infection. By binding to the three structural proteins GP2, GP4 and GP5, CD163 mediates entry and attachment of the virus to susceptible host cells [56]. 

Since macrophages polarization plays an important role in PRRSV infection, we then characterized M1/M2 macrophages polarization on PAMs challenged by A1/A2. Interestingly, our results showed that A1 potently repolarizes macrophages into M1 macrophages compared to macrophages challenged with A2, as indicated by significant upregulation of pro-inflammatory genes (*TNF-α, IL-6* and *IL-12*) in PAMs challenged with A1 (Figure 2a). In contrast, A2 induction significantly up-regulates the anti-inflammatory gene *Arg-1*. Moreover, data from next-generation sequencing (NGS) supported our gene profiling results. They revealed that some pro-inflammatory genes were upregulated in PAMs challenged with A1 compared to A2 (Figure 3a,b). These genes include *Nf-kβ, NNT, NFAT5, PI3KCB,* and *JARID2*. Nuclear factor-κβ (Nf-kβ) synthesizes cytokines, such as *TNF-α*, IL-1β, *IL-6*, and IL-8, and thereby regulates pro-inflammatory genes [57]. The other pro-inflammatory genes, *nuclear factor of activated T-cells 5* (*NFAT5*), enhance pro-inflammatory macrophage polarization when appropriately stimulated and would be neutralized in a pro-M2 tumor microenvironment [58,59,60]. Taken together, the NGS reported the shared genes involved in regulating signaling pathway. Additionally, we found that the T-cells receptor (TCR) signaling pathway were up-regulated (Figure 3d). A study demonstrated that the strength of TCR signaling guides the development of CD8^+^ T-cells in the thymus, a process that may have a direct impact on autoimmune diseases [61]. In cooperation with cytokine signaling pathways, co-stimulatory molecules, chemokines, integrins and metabolites, TCR signaling drives the differentiation of activated T-cells into specific T-cell subtypes, namely T helper type 1 (Th1), Th2, Th17, follicular T helper and regulatory T-cells (Treg) [62]. 

In order to investigate the relevance of the different sets of genes, we created the KEGG Pathway Enrichment Analysis. Our analysis showed the enrichment of a total of 19 KEGG and 4 KEGG pathways in 50 up- and down-regulated genes in A1-challenged PAMs compared to their counterpart challenged by A2 (Figure 4 and Figure 5). KEGG pathway enrichment analysis revealed that the repressor activator protein 1 (Rap1) pathway is one of the most highlighted pathways and is modulated by TCR activation, supporting our previous finding from NGS. Rap1 plays a critical role in modulating T-cell responses and regulates interactions between T-cells and antigen-presenting cells (APCs). In one study, antigen-dependent Rap1 activation was shown to enhance T-cell-APC interactions and trigger activation-induced cell death (AICD) [63]. Additionally, the down-regulation of anti-inflammatory genes has significant correlation with estrogen signaling pathway. Estrogen functions as a key regulator of anti-inflammatory responses and stimulates alternative macrophages polarization during cutaneous repair [64]. These results strongly suggest that A1 promotes the repolarization of PAMs toward M1 and can regulate the activation of Th1 cells through the secretion of pro-inflammatory cytokines.

The defenses of the innate immune system depend on three complement pathways in triggering a localized inflammatory response (which are the classical pathway, the alternative or properdin pathway, and the lectin pathway) to initiate a localized inflammatory response. Activation of the lectin pathway was not dependent on antibodies and is caused by the attachment of plasma mannose-binding lectin (MBL) to microbes [65]. In line with the reference, we found that C-type lectin receptors (CLRs) pathway was enriched in A1-challenged PAMs. CLRs signaling could be one of the targets for vaccine development. Signaling pathways induced by CLRs have the potential to directly activate the transcription factor NF-κB. Recognition of CLRs leads to several properties that are important for vaccine design, including pathogen internalization, degradation, and subsequent antigen presentation [66].

We then tested the effect of recombinant antigens on Th1 cells activation. We used a T-cell co-culture system for the experiments, and the results showed that A1 promoted Th1 activation by secretion of IFN-γ cytokines (Figure 6b). Our results are in agreement with the literature showing that the inflammatory process is dominated by IFN-γ-producing Th1 cells [67]. It is known that activated Th1 cells potentially secrete proinflammatory cytokines IL-2, IFN-γ and lymphotoxin-α (LT, TNF-β), and enhance B-cells maturation and the production of IgG2a antibody that optimize clearance of viruses and extracellular bacteria [29]. Moreover, Th1 cells are involved in cell-mediated immunity (CMI) by inhibiting infections through induction of complement fixation, opsonizing antibodies, and antibodies involved in antibody-dependent cell cytotoxicity, such as IgG1 in humans and IgG2a in mice [68]. CMI is clearly one of the most important regulators in PRRSV infection [69]. Gujer et al., 2011, [70] determined that IFN-γ secreted by human plasmacytoid dendritic cells (PDCs) stimulate B-cells proliferation and differentiation to antibody-secreting cells by enhancing the interactions of B- cells and T-cells. Surprisingly, the pro-inflammatory gene profile of *TNF-α* was boosted up to 80-fold after A1-induced PAMs interacted with Th1 cells when we compared it with A1-induced PAMs alone in the mono cell culture system (only 15-fold) (Figure 6e). In combination of IFN-γ secretion, these results proved that Th1 cell activation was triggered by cell-cell interaction of A1-induced PAMs and T cells. In addition, the expression of *IL-6* was also significantly increased after being challenged with A1 compared to the control and A2 groups (Figure 6d). It is noteworthy that *IL-6* binds to the receptors (CD126 and CD130) to regulate B-cells proliferation [71]. Furthermore, our results indicate that A1 might regulate the Th1 type response through the interaction of the Caspase-1 (Casp1) and interleukin 18 (IL-18) proteins. Pro-inflammatory cytokine IL-18 has the potent ability to support Th1 activation, but not Th1 development, and secrete IFN-γ in the presence of *IL-12* [72]. Indirectly, the up-regulated gene sets promote the interaction of inhibitor of the nuclear factor kappa B kinase subunit B (IKBKB or IKKβ) protein, thus promote B-cells differentiation. IKKβ is considered to be a key factor of antibody-mediated immunity that provides an essential role in the activation of B-cells and the maintenance of B-cell survival, and IKKβ knockdown significantly induce cell death in all peripheral B-cell [73]. Besides, we described that the down-regulated genes in PAMs challenged by A1 possibly inhibits the interaction of ISG15 and USP18 as negative regulators of the immune response. High levels of free ISG15 were confirmed to be released by M2 macrophages and exert pro-tumor activity in pancreatic ductal adenocarcinoma (PDAC) by regulating PD-L1 expression [74]. By directly inhibiting type I IFN receptor signaling and regulating the expression program cell death-1, Usp18 has been shown to function as a negative regulator of innate anti-viral responses [75]. 

The question was, unlike A1, why A2 did not provoke Th1 response. One possible explanation for it was that A2 was engineered with lipidated recombinant protein that provides support only for B-cells activation as a humoral immune response, while A1 was engineered with T-cell epitopes that promote cell-mediated immune response. T-cell-inducing vaccines have been developed to induce CD4^+^ and/or CD8^+^ cells to give a protective adaptive immune response against viral infection. Recent clinical trials have now reported the strong protective effect of T-cell-inducing vaccines against a range of diseases, including HIV, malaria, influenza, tuberculosis, and cancer [76]. On the contrary, vaccination of mice with lipidated pneumococcal lipoproteins enhanced antibody-mediated immunity compared with vaccination with non-lipidated proteins, but had no significant effect on the CMI responses [77]. Another possibility is caused by *ORF7*, which is involved in A2. Conceivably, it is possible that *ORF7* should play an essential role in inhibiting the cell-mediated immune response in PAMs against PRRSV. As described in a previous study, a robust T-cell response was induced by the PRRSV *ORF6* (M) protein, whereas the PRRSV *ORF7* protein induced a weak ability to stimulate T-cells [78]. Similarly, the neutralizing antibodies were not generated by antibodies directed against *ORF7*, although this is the most abundant and used for serological tests [79,80]. Another study compared the two vaccines developed, namely the PRRSV-*ORF7* DNA vaccine (phCMV-*ORF7*) alone and PRRSV-*ORF7* with IL-2 as an adjuvant. The results showed that PRRSV-*ORF7* DNA vaccine (phCMV-*ORF7*) alone was insufficient to induce protective immunity, but had a positive inductive effect on activating vaccine-induced virus-specific cellular immunity when IL-2 was added [81]. However, the role of cell-mediated immunity to *ORF7* in protection against PRRSV is still controversial. Finally, we conclude that PRRSV-2 recombinant protein antigen A1 was more potent than A2 in protecting PRRSV infection by inducing repolarization of M1 macrophages and suppressing CD163 receptor expression for virus entry. Similarly, PRRSV-2 recombinant antigen A1 enhances the immune response by stimulating endogenous genes to activate the Th1 type immune response, and potentially facilitates B-cells proliferation and differentiation into antibody-secreting cells. Overall, our new findings could be useful for the development of screening new vaccine candidates against PRRSV. In addition, future studies including immunization and humoral immune response analyses are needed for a better understanding of long-lasting protection and the development of a potential vaccine against PRRSV.

## 5. Conclusions

Based on the results of our study, recombinant PRRSV-2 antigen A1, which consists of the complete sequence of *ORF5*, a partial sequence of *ORF6*, and T-cell epitopes, can stimulate the repolarization of M2 PAMs to M1. These repolarized M1 PAMs decreased the total expression of CD163 for PRRSV entry, and hence offer a broad protection for both PRRSV-1 and PRRSV-2 strains infection. Cell-cell interaction of A1-induced PAMs and T cells can further stimulate Th1 response, along with the activation of T cell receptor signaling pathway in PAMs and the secretion of IFN-γ from T cells, respectively. These results also suggest that macrophages are not only host target cells but also play a key role in immunomodulation during PRRSV infections. In addition to humoral immunity, our assays provide a vision to screening the antigenicity of novel subunit-protein vaccine candidates for their innate and T-cells-mediated immunity. 

## Figures and Tables

**Figure 1 vaccines-09-01009-f001:**
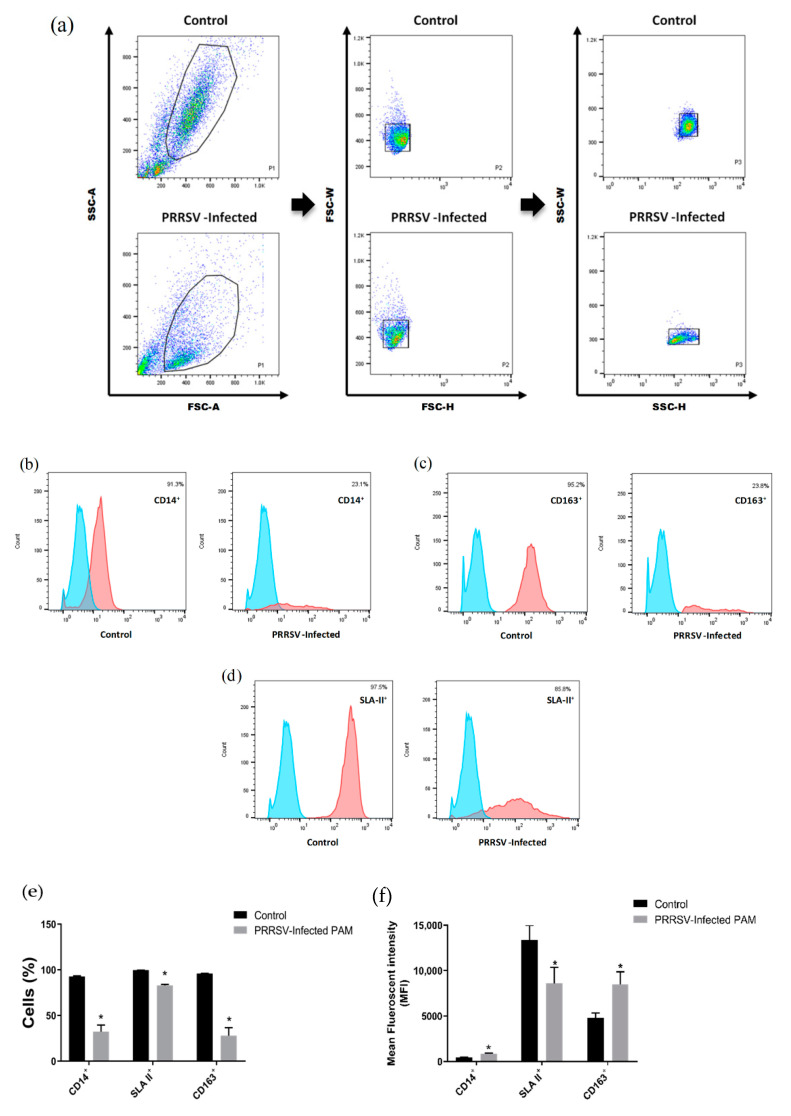
Surface marker expression on alveolar macrophages from animals infected with PRRSV. (**a**) Gating strategy of healthy PAMs and PRRSV-Infected PAMS. Dot plots (FSC-Area vs. SSC-Area) from a representative pig of the control and PRRSV-infected group. We excluded more dead cells in the lower levels of FSC/SSC. The circles indicate living potential of PAMs according to light scatter properties (size and granularity). The combination of width and height of FSC and SSC, respectively, indicate the singlets of cells. (**b**–**d**) Representative flow cytometry histogram of PRRSV-infected PAMs. Samples were calculated from three pigs (duplicate each pig). Red color histogram represents signal for each specific antibody, and blue color histogram represents unstained. Cells percentage (**e**,**f**) mean fluorescence intensity (MFI) of CD14^+^, SLA II^+^ and CD163^+^ were detected with a specific monoclonal antibody, followed by a FITC-conjugated anti-mouse antibody and analysis by flow cytometry. The level of expression of non-infected animals is included as a control. Statistical differences were established by mean ± SEM (* *p* < 0.05).

**Figure 2 vaccines-09-01009-f002:**
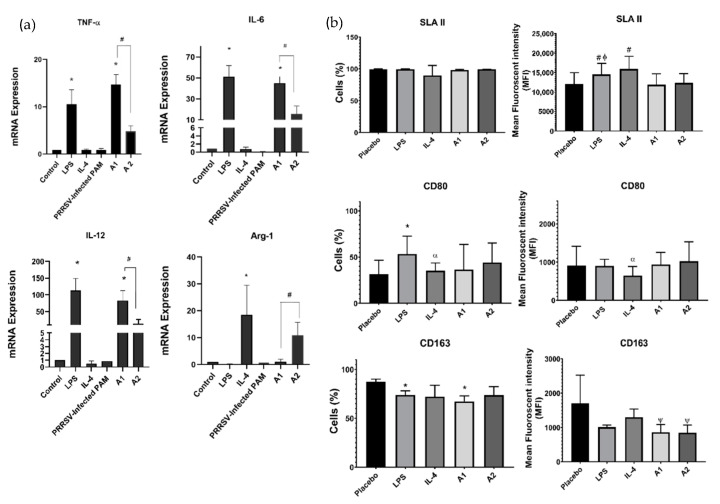
A1 drives M1 macrophages polarization and the expression of CD163 on PAMs was significantly decreased. (**a**) mRNA expression profile of M1 and M2 macrophages of PAMs-challenged by PRRSV-2 antigens measured by quantitative PCR (qPCR). A1 promote up-regulation the mRNA level of pro-inflammatory genes and down-regulate the anti-inflammatory genes. (**b**) The expressions of surface protein SLA II^+^, CD80^+^ and CD163^+^ in macrophages by flow cytometry. A1 decreased the cells percentage of CD163^+^ on PAMs. PAMs were treated by LPS, IL-4, A1, or A2 for 24 h. Untreated PAMs were used as control. * *p* < 0.05 compared to control group; α *p* < 0.05 compared to LPS group; ψ *p* < 0.05 compared to IL-4 group; # *p* < 0.05 compared to A1 group; ϕ *p* < 0.05 compared to A2 group. The data were analyzed by mean ± SEM, calculated from 3 pigs (duplicate each pig).

**Figure 3 vaccines-09-01009-f003:**
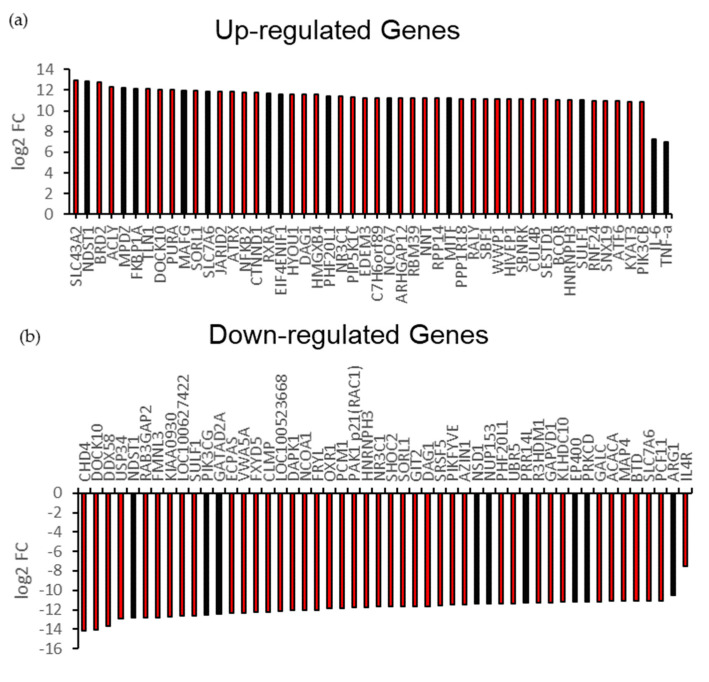
Transcriptome Profiling by Next Generation Sequencing in PAMs challenged with Antigens. Top 50 (**a**) up-regulated and (**b**) down-regulated differentially expressed genes in PAMs challenged by A1 compared to A2. The highlighted bars indicate the same pattern of the expressed genes while A1 was compared to the control. (**c**) Venn diagram based on the overlapping differentially expressed genes in PAMs challenged by A1 compared to A2. (**d**) Up-regulated and (**e**) down-regulated pathways stimulated by A1. The T-cell receptor signaling pathways were up-regulated on A1 challenged.

**Figure 4 vaccines-09-01009-f004:**
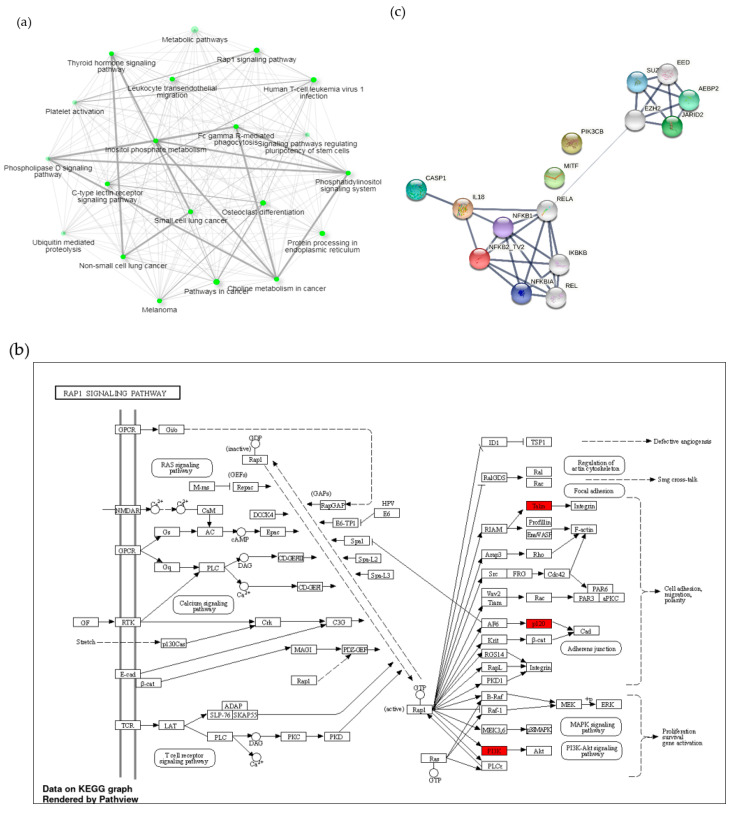
Network displaying the predicted interaction of signaling pathways and cytokine-related genes involved in immune response process. (**a**) Relationship between enriched pathways enhanced by up-regulated genes by A1 induction. Nodes represent each pathway; two nodes are connected if they share 20% or more genes. Darker nodes are more significantly enriched gene sets. Bigger nodes indicate larger gene sets. Thicker edges represent more overlapped genes. (**b**) Rap1 signaling pathways mediate TCR signaling pathway activation, thus promote proliferation, survival, and gene activation. Three out of 50 up-regulated genes are represented by the red box. (**c**) Protein-protein interactions involved in the activity of the immune response. Nodes with different colors represent different enriched proteins.

**Figure 5 vaccines-09-01009-f005:**
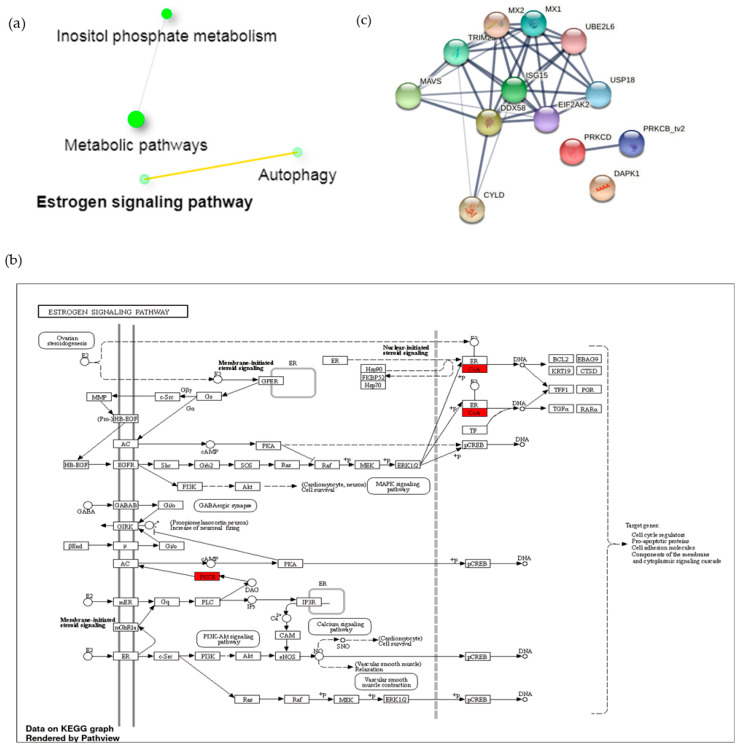
Network displaying the predicted interaction of signaling pathways and cytokine-related genes affected by suppression of anti-inflammatory genes. (**a**) Relationship between enriched pathways enhanced by down-regulated genes in A1-challenged. Nodes represent each pathway; two nodes are connected if they share 20% or more genes. Darker nodes are more significantly enriched gene sets. Bigger nodes indicate larger gene sets. Thicker edges represent more overlapped genes. (**b**) Estrogen signaling pathways and (**c**) protein-protein interactions showing predicted interactions between proteins encoded by down-regulated differentially expressed genes. Nodes with different colors represent different enriched proteins. Filled nodes indicate that a 3D structure is predicted. Edges represent protein-protein association or contribute a shared function.

**Figure 6 vaccines-09-01009-f006:**
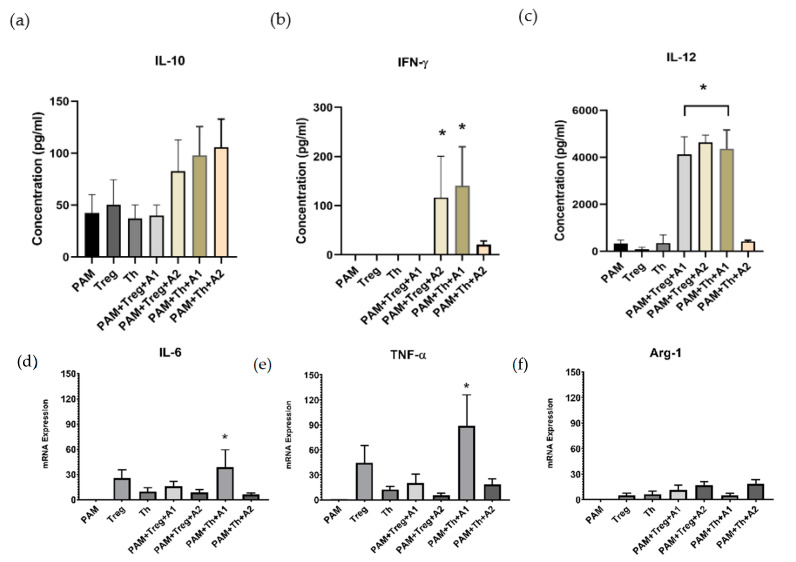
PAMs induced by A1 Activate Th1 cells. The conditioned medium was collected from PAMs on 48 h post-challenge and analyzed by ELISA to determine cytokine levels: (**a**) IL-10, (**b**) IFN-γ, (**c**) *IL-12*. Each bar represents the number of cytokines secreted by individual samples. Pro-inflammatory gene expression determined by real-time PCR. The genes profile of (**d**) *IL-6*, (**e**) *TNF-α* were up-regulated by challenge with A1, but no any significant at (**f**) *Arg-1*. Isolated PAMs were challenged with antigens followed by co-cultured with CD4^+^CD25^+^ Tregs or CD4^+^CD25^−^ Th1 cells for 48 h. Protein concentrations were measured by ELISA and determined by mean ± SEM, calculated from 3 pigs (duplicate each pig), * *p* < 0.05 compared to PAMs group.

**Table 1 vaccines-09-01009-t001:** Primer sequences.

Gene Name	Primer Sequences (5′–3′)
Porcine *IL-6*	GCTGCTTCTGGTGATGGCTACTGCC
	TGAAACTCCACAAGACCGGTGGTGA
Porcine *TNF-α*	ATGAGCACTGAGAGCATGATCCG
	CCTCGAAGTGCAGTAGGCAGA
Porcine *Arg-1*	AGCCCGTGTCAACATGACTTCC
	TTGTGTTGGCATCTTTACTGA
Porcine *IL-12*	CTCCCACACCGAAGCTTGAA
	TTCTTCACCATGGGGGCT
Porcine *β-actin*	ACAGACAGCCGTGTGTTCC
	ACCTTCACCATCGTGTCTCA

## Data Availability

Data are contained within the article. Reported results can be found in Appendix A.

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
