# Peer review of "Recombinant Antigen of Type 2 Porcine Reproductive and Respiratory Syndrome Virus (PRRSV-2) Promotes M1 Repolarization of Porcine Alveolar Macrophages and Th1 Type Response"

_vaccines, 2021, doi:10.3390/vaccines9091009_

Round 1

Reviewer 1 Report

The manuscript by Wahyuningtyas et al. reported porcine alveolar macrophage (PAM) response to the stimulation by porcine reproductive and respiratory syndrome virus (PRRSV) protein A1 (g6Ld10T) and A2 (lipo-32 M5Nt). Using flow cytometry, the authors characterized PAM by specific cell markers and assayed the gene transcription activation and cytokine expression level by the stimulation of A1 and A2. The authors discovered pro-inflammation activation and reduced expression of a PRRSV host receptor, CD163, in these cells by A1 stimulation but not A2, which the authors claimed as a potential implication of A1 vaccination to prevent PRRSV infection. While the data presented here is of general interest to the PRRSV study, several concerns should be addressed before further consideration.

  1. It is not clear how CD14 and CD163 expression level was up-regulated in PRRSV infection as shown in Fig 1b.
  2. There is lacking of control group when conducting transcriptome analysis. It is not clear whether the apparent “up-regulation” was resulted from increased transcription in A1 group or decreased transcription in A2 group. Comparison with mock treated PAM cells should be performed. Lacking of this data may impact the conclusion in the manuscript.
  3. Fig 2b bottom right panel, not sure how IL-4 is compared with IL-4 group with a P<0.05.
  4. Fig 6b should be labeled as “IFN-γ” according to the term used in main text.
  5. It is not clear what the negative concentration in Fig 6b can tell us.
  6. No supportive data on whether A1 vaccination increases the neutralization potency of sera in vaccinated animals.

Reviewer 2 Report

The manuscript titled “Recombinant Antigen of Type 1 Porcine Reproductive and Respiratory Syndrome Virus (PRRSV-1) Promotes M1 Repolarization of Porcine Alveolar Macrophages and Th1 Type Response” from Wahyuningtyas et al. shows the effect of two PRRRSV-1 antigens on macrophage polarization. The article is well written and I have only several minor comments that need to be addressed:

  • Figure 1: Add the gating strategy (FSC/SSC). The flow cytometry dot plots need to be presented in bi-exponential mode.
  • The authors did not exclude the dead cells with a DAPI staining or any other dead/live cell stainings during the flow cytometry acquisition. How did the authors control unspecific antibody binding due to dead cells during viral infection?
  • The resolution of Figure 5b is too low to be able to read it clearly.
  • The authors need to use the same y-axis scale for Figure 6 d, e and f.

Round 2

Reviewer 1 Report

The authors have added limited data but did not address my concerns in depth. Several critical points determining the scientific soundness have not been well resolved.

  1. The gene expression profiles have not been properly analyzed. The heatmap comparison data are confusing. For instance, TLR8 and UTRN are up-regulated in both A1 and A2 group but interpreted by the authors as both up-regulated and down-regulated at the same time in the A1 vs A2 heatmap (Figure S3). Most importantly, the authors do not show how the internal control of the gene expression profile and data normalization been performed in order to give a consistent comparison. The number of replicates are also critical information and should be provided in the legend. Moreover, there is lacking of description section in the manuscript about the internal control, normalization, comparison algorithms (not the name of R package) of these gene expression profiling.
  2. The negative contractions reported in Figure 6b are not meaningful but reflecting an improper experiment setup. The concentration numbers should reflect the real concentration in medium. A standard curve usually titrated with a serial diluted concentration of standard interferons can be used to determine a range of concentrations falling in the linear range of the standard curve. The author should be responsible for setting up the standard curve properly including titrating an appropriate range of concentrations to meet the experiment requirement.
  3. The authors did not make changes on Figure 2b although they addressed a revision has been made in the figure.

Round 3

Reviewer 1 Report

I appreciate the authors for their efforts to address all my previous concerns. I have no more concerns except Figure 6f is missing in the revised manuscript.
